# Blood Pressure Change from Normal to 2017 ACC/AHA Defined Stage 1 Hypertension and Cardiovascular Risk

**DOI:** 10.3390/jcm8060820

**Published:** 2019-06-08

**Authors:** Joung Sik Son, Seulggie Choi, Gyeongsil Lee, Su-Min Jeong, Sung Min Kim, Kyuwoong Kim, Jae Moon Yun, Sang Min Park

**Affiliations:** 1Department of Family Medicine, Seoul National University Hospital, Seoul 03080, Korea; medical114@naver.com (J.S.S.); gespino1.gs@gmail.com (G.L.); dpsme@naver.com (S.-M.J.); jaemoon2@gmail.com (J.M.Y.); 2Department of Biomedical Sciences, Seoul National University Graduate School, Seoul 03080, Korea; seulggie@gmail.com (S.C.); ksm9904@naver.com (S.M.K.); kwkim238@gmail.com (K.K.)

**Keywords:** blood pressure change, 2017 ACC/AHA high blood pressure guideline, cardiovascular disease, stage 1 hypertension

## Abstract

The purpose of this study was to investigate the clinical significance of the 2017 American College of Cardiology (ACC)/American Heart Association (AHA) defined stage 1 hypertension (systolic blood pressure (SBP) 130–139 mmHg or diastolic blood pressure (DBP) 80–89 mmHg), and increase in BP from previously normal BP in Korean adults. We conducted a retrospective analysis of 60,866 participants from a nationally representative claims database. Study subjects had normal BP (SBP < 120 mmHg and DBP < 80 mmHg), no history of anti-hypertensive medication, and cardiovascular disease (CVD) in the first period (2002–2003). The BP change was defined according to the BP difference between the first and second period (2004–2005). We used time-dependent Cox proportional hazards models in order to evaluate the effect of BP elevation on mortality and CVD with a mean follow-up of 7.8 years. Compared to those who maintained normal BP during the second period, participants with BP elevation from normal BP to stage 1 hypertension had a higher risk for CVD (adjusted hazard ratio (aHR) 1.23; 95% confidence interval (CI), 1.08–1.40), and ischemic stroke (aHR 1.32; 95% CI, 1.06–1.64). BP elevation to 2017 ACC/AHA defined elevated BP (SBP 120–129 mmHg and DBP < 80 mmHg) was associated with an increased risk of CVD (aHR 1.26; 95% CI, 1.06–1.50), but stage 1 isolated diastolic hypertension (SBP < 130 and DBP 80–89 mmHg) was not significantly related with CVD risk (aHR 1.12; 95% CI, 0.95–1.31).

## 1. Introduction

Hypertension is a modifiable risk factor for cardiovascular disease (CVD) [1,2]. The population-attributable fraction of hypertension for CVD is up to 60% in the Asia-Pacific region [3,4]. The linear relationship between blood pressure (BP) and the risk of CVD has been well demonstrated. A meta-analysis of 61 prospective studies demonstrated that mortality from ischemic heart disease and stroke increases linearly from levels as low as systolic BP (SBP) 115 mmHg and diastolic BP (DBP) 75 mmHg [5]. In the seventh report of the Joint National Committee on Prevention, Detection, Evaluation, and Treatment of High Blood Pressure (JNC 7), a new BP category, prehypertension (SBP 120–139 mmHg or DBP 80–89 mmHg) was introduced [6]. Recently, meta-analysis studies have shown that prehypertension is associated with a high risk of CVD [7,8].

In 2017, the American College of Cardiology (ACC) and the American Heart Association (AHA) released a new guideline on hypertension with a new definition of stage 1 hypertension (SBP 130–139 mmHg or DBP 80–89 mm) [9]. Two cohort studies have shown that stage 1 hypertension in young adults was associated with an increased risk of subsequent CVD. [10,11] Furthermore, changes and variability in BP have also been suggested as independent risk factors for CVD [12,13,14]. Prospective observational cohort studies have indicated that changes in BP were associated with the risk of cardiovascular events and CVD related mortality [15,16]. Meanwhile, studies investigating of the effect of BP changes among those with normal BP on CVD and mortality are lacking, especially upon BP elevation to 2017 ACC/AHA defined Stage 1 hypertension (Stage 1 hypertension). In the present study, we aimed to investigate the relationship between changes in BP and the risk of CVD and death in people with normal BP.

## 2. Materials and Methods

### 2.1. Study Population

The current study used the National Health Insurance Service-National Health Screening Cohort (NHIS-HEALS), which is a cohort randomly selected among those who underwent national health screening from the Korean population by the NHIS in South Korea. The NHIS database includes many health check-up items based on anthropometric measurements, physical examinations, and health examinees’ questionnaire results. In addition, blood pressure measurement and laboratory tests were performed, including complete blood count (CBC), fasting serum glucose (FSG), total cholesterol, and dipstick urine tests (occult blood, glucose and protein). Among the participants aged 40–79 years who participated in the biennial national health screening program covered in the NHIS in 2002 and 2003, 10% of the participants were randomly selected. This cohort was followed up for 12 years until 2013.

We identified 334,632 participants who had BP values in the first (2002–2003) and second (2004–2005) health check-up periods. We excluded those who passed away (*n* = 1043) before the index date of 1 January 2006. In addition, we also excluded those who had previous medical histories of myocardial infarction (MI) or stroke by using the Tenth Revision of the International Classification of Diseases (ICD-10) (I20–I25 for MI and I60–I69 for stroke) diagnosis codes, and a self-reported questionnaire on histories of MI and stroke before the index date (*n* = 38,825). We excluded subjects with SBP greater than 120 mmHg or DBP greater than 80 mmHg in the first (2002–2003) period (*n* = 218,626), and subjects with a self-reported previous history of hypertension and antihypertensive medication (*n* = 14,019), or who had missing values on laboratory results and lifestyle variables at the health check-up (*n* = 3300), resulting in a total of 60,866 participants (Figure 1). This study was approved by Seoul National University Hospital’s Institutional Review Board (IRB) (IRB number: 1703-039-863), and consent from individual patients was waived as NHIS-HEALS is de-identified according to strict confidentiality guidelines.

### 2.2. Change in BP Level

Resting BP was measured after participants rested for at least 5 minutes in sitting position by digital or automatic BP monitors at each health check-up. BP measured during the second period (2004–2005) was divided into four groups: normal (SBP < 120 mmHg and DBP < 80 mmHg), 2017 ACC/AHA defined elevated BP (Elevated BP) (SBP 120–129 mmHg and DBP < 80 mmHg), Stage 1 hypertension (SBP 130–139 mmHg or DBP 80–89 mmHg) and Stage 2 hypertension (SBP ≥ 140 mmHg or DBP ≥ 90 mmHg). For further analysis, the Stage 1 hypertension participants were divided into Stage 1 isolated diastolic hypertension (IDH) (SBP < 130 DBP 80–89 mmHg), Stage 1 isolated systolic hypertension (ISH) (SBP 130–139 DBP < 80 mmHg), and Stage 1 combined systolic/diastolic hypertension (SDH) (SBP 130–139 DBP 80–89 mmHg) groups.

### 2.3. Follow-up and Outcome Measurement

The mean follow-up period was 7.8 years. Deaths among study population through the end of follow-up date, 31 December 2013 were confirmed by using death certificates from Statistics Korea. CVD mortality (I00–I99) was identified by codes of death. The primary outcome was CVD. A CVD event was defined as a composite end point of two or more days of hospitalization due to acute myocardial infarction (AMI) (I21), stroke (I60–I69) or CVD mortality (I00–I99) according to the International Classification of Diseases, Tenth Revision (ICD-10) codes. The secondary outcomes were AMI (I21), ischemic stroke (I63), hemorrhagic stroke (I60–I62), all-cause mortality and CVD mortality.

### 2.4. Statistical Analysis

The follow-up period began on 1 January 2006 and extended through 31 December 2013. Each participant was censored at the time of CVD event or deaths from any causes, whichever occurred first. In case of non-censored observations, the participants were followed-up until 31 December 2013. We used Cox proportional hazards regression models in order to evaluate mortality and CVD events according to the change in BP levels. Age was divided into four categories, 40–49, 50–59, 60–69, and ≥70 years (yr). Body mass index (BMI) was calculated by dividing the weight in kilograms by the height in meters squared and classified into <18.5, 18.5–22.9, 23.0–24.9, and ≥25.0 kg/m^2^ according to the Asian-pacific obesity classification [17]. Comorbidities were summarized by the Charlson comorbidity index (CCI) with ICD-10 coding, as described elsewhere [18]. The use of aspirin was defined as subjects who had been prescribed aspirin between 2002 and 2005. Mean prescribed daily doses for statin were converted to defined daily dose (DDD) and subjects who have claim record more than 30 cumulative DDDs between 2002 and 2005 were defined as statin users [19,20].

We adjusted for age and sex in model 1. We additionally adjusted for CVD risk factors including body mass index (<18.5, 18.5–22.9, 23.0–24.9, ≥25.0 kg/m^2^), fasting serum glucose (<100.0 mg/dL, 100–125.9 mg/dL, ≥126.0 mg/dL), total cholesterol (<200.0 mg/dL 200.0–239.9 mg/dL, ≥240.0 mg/dL), physical activity (none, 1–2 times/week, 3–4 times/week, and 5–7 times/week), smoking status (Never smoker, Ex-smoker, Current smoker), drinking habit (none, 1–2 times/week, ≥3 times/week), socioeconomic status (Lower half, Upper half, Medical Aid), Charlson comorbidity index (0, 1–2, and ≥3), use of aspirin, and use of statins in order to estimate hazard ratios (HR) and a 95% confidence interval (95% CI) among Elevated BP (SBP 120–129 mmHg and DBP < 80 mmHg), Stage 1 hypertension (SBP 130–139 mmHg or DBP 80–89 mmHg) and Stage 2 hypertension (SBP ≥ 140 mmHg or DBP ≥ 90 mmHg) as compared to those with normal BP (SBP < 120 mmHg and DBP < 80 mmHg). We performed subgroup analysis by stratifying the study sample by sex, baseline age (40–64 and ≥65 yr), fasting serum glucose level (<126.0 mg/dL, ≥126.0 mg/dL), smoking status (Never smoker, Ex- or Current smoker) and Charlson comorbidity index (0–1, and ≥2). Data management and collection were carried out using SAS version 9.4 (SAS Inc, Cary, NC, USA). All statistical analyses were carried out using STATA version 14.1 (Stata Corp, College Station, TX, USA). All tests were two-sided, and statistical significance was defined as a *p* value of less than 0.05.

## 3. Results

### 3.1. Baseline Characteristics

The mean age of the total population was 51.5 (standard deviation 7.6) years old and were composed of 52.5% female subjects (Table 1). The subjects were distributed into 32,583 participants (53.5%) in the normal BP group, 6702 participants (11.0%) in the Elevated BP group, 16,880 participants (27.7%) in the Stage 1 hypertension group, and 4701 participants (7.7%) in the Stage 2 hypertension group. The incidence for all-cause mortality, CVD mortality, CVD, myocardial infarction, ischemic stroke, and hemorrhagic stroke were 1517 (2.49%), 165 (0.27%), 1285 (2.11%), 156 (0.26%), 470 (0.77%) and 198 (0.33%), respectively. Subjects in the hypertension group were more likely to be old, male, obese, and current smokers. Those in the hypertension group showed higher total cholesterol and fasting serum glucose levels compared to those in the normal BP group. Upon dividing the Stage 1 hypertension groups into Stage 1 IDH, ISH, and SDH, the mean ages were 51.0, 54.4, and 52.7 years, respectively.

### 3.2. Change in BP and Mortality and Cardiovascular Disease

Participants with blood pressure elevation from normal BP to newly defined Stage 1 hypertension had higher risk for CVD (aHR 1.23; 95% CI, 1.08–1.40), and ischemic stroke (aHR 1.32; 95% CI, 1.06–1.64) than those who maintained normal BP (Table 2) (Figure 2). People whose BP rose up from normal BP to the Elevated BP had higher risk for CVD (aHR 1.26; 95% CI, 1.06–1.50), myocardial infarction (aHR 1.91; 95% CI, 1.19–3.06), and ischemic stroke (aHR 1.38; 95% CI, 1.04–1.84) compared to those who maintained normal BP (Table 2) (Figure 2). Compared to those who maintained normal BP, those who had higher BP to Stage 1 IDH levels did not have increased risk for CVD (aHR 1.12; 95% CI, 0.95–1.31) (Table 3). In contrast, participants with Stage 1 SDH (aHR 1.42; 95% CI 1.17–1.71) levels during the second period had elevated risk for CVD. Compared to those with Stage 1 IDH, people with Stage 1 SDH had higher CVD risk (aHR 1.29, 95% CI, 1.04–1.61) (Table 4). The association of BP elevation with CVD was consistent in several subgroups (Table 5).

## 4. Discussion

In people with normal BP, BP elevation within two years was associated with an increased risk of CVD. Increases in BP to Stage 1 hypertension levels were associated with increased risk of CVD. BP elevation to Elevated BP level was also associated with an increased risk of CVD, but Stage 1 IDH was not significantly related.

Previous research by Hennekens CH et al. on a two-year BP change and the risk of CVD in men reported that a two-year change in DBP, but not SBP, may be associated with CVD risk [16]. The study of Hennekens CH et al. showed that SBP differences above 15 mmHg were not associated with additional increase in CVD risk. However, Hadaegh F et al. reported that three-year rises in SBP (≥ 13.8 mmHg), DBP (≥9.14 mmHg) were associated with increased risk of CVD [21]. This difference seems to be due to the racial difference (US, Iran, and Asian) in the study subjects, participation of subjects with BP above normal BP, the differences in the observation periods, and the adjusted variables [22].

In an analysis including a total of 346,570 participants from 36 Asia-Pacific region cohort studies, JNC 7 defined prehypertension (SBP 120-139 mmHg or DBP 80–89 mmHg) was associated with increased risks of coronary heart disease (aHR 1.31; 95% CI, 1.14-1.50), ischemic stroke (aHR 1.60; 95% CI, 1.33–1.92) and hemorrhagic stroke (aHR 2.17; 95% CI, 1.69–2.79) [7]. Ishikawa S et al. reported that the risk of CVD with prehypertension might increase after progression to hypertension. However, increased BP from normal BP to prehypertension was not associated with CVD [23]. The two-year changes in BP may be due to changes in behavior, medication usage, or regression to the mean.

The 2017 ACC/AHA High Blood Pressure Guideline introduced a new definition of Stage 1 hypertension (SBP 130 to 139 mmHg or DBP 80 to 89 mmHg) [9]. Therefore, some of those previously diagnosed with prehypertension would be diagnosed with Stage 1 hypertension. The results of this study are of public health value that emphasizes the elevated risk of CVD in Elevated BP levels among previously normal BP individuals.

There did not appear to be a significant difference in the risk of CVD between Elevated BP group and Stage 1 hypertension group (Table 2) (Figure 2). Additional analysis showed that the Stage 1 hypertension group could be classified as a heterogeneous group: IDH, ISH, and SDH. In particular, people with Stage 1 IDH accounted for 62% of the Stage 1 hypertension group and their CVD risk was not statistical significantly elevated compared to those who maintained normal BP (Table 3). Despite being categorized as the same Stage 1 hypertension group, people with Stage 1 SDH had higher CVD risk than those with Stage 1 IDH (aHR 1.29, 95% CI, 1.04–1.61) (Table 4). The study population consisted of participants who had normal BP during the first visit, and Stage 1 IDH patients had average SBP and DBP values of 118.1 ± 5.6 and 80.7 ± 1.8 mmHg, respectively. Therefore, as the mean SBP values were below 120 mmHg, the risk for CVD did not seem to increase. Moreover, as the Stage 1 IDH group had a mean age of 51.0 years, the CVD risk may not have increased compared to that of Stage 1 ISH and SDH participants, who had mean age of 54.4 and 52.7 years, respectively. In a study of elderly hypertensive patients, the risk of CVD was similar between SDH and ISH [24]. Further, a long-term follow-up study of young and middle-aged adults showed that ISH was associated with CVD risk [25]. However, there was no statistically significant correlation between IDH and stroke in a study of middle-aged and elderly [26]. Further research is needed on the clinical significance and within-group heterogeneity of the Stage 1 hypertension group.

The limitations of this study are as follows. First, there is a methodological limitation in BP measurements. The hypertension guidelines recommend that BP be measured repeatedly and averaged, but BP measurements in our study were conducted only once. The study of Handler J et al. showed that one BP measurement was appropriate for people with baseline normal BP [27]. However, a cross-sectional study revealed that there were significant differences between a single office BP measurement and the mean of consecutive BP measurements. Therefore, a short-term masked hypertension might not be found using only one BP measurement value [28]. Second, we could not consider the possible effects of medications on the relationship between BP change and CVD during the follow-up period. Further studies are needed to analyze the relationship between drug use during the follow-up period and changes in BP and the incidence of CVD. Third, the study subjects were Koreans; therefore, there is a limit to the generalization of research results to other countries or races.

There are also several strengths. To the best of our knowledge, this is the first study to show that the CVD risk increased upon BP elevation from normal BP among those with two-year BP change from normal BP to elevated BP and Stage 1 hypertension. The current study is a relatively large study population that is representative of the general population with a relatively long-term follow-up duration. This study also considered a number of potential confounders such as drug prescription, sociodemographics, and health behaviors.

## 5. Conclusions

In conclusion, the results from the current study revealed that BP elevation within two years was associated with an elevated risk of CVD among those with baseline normal BP. BP elevation to Stage 1 hypertension and Elevated BP was also associated with an increased risk of CVD, but Stage 1 IDH was not significantly related.

## Figures and Tables

**Figure 1 jcm-08-00820-f001:**
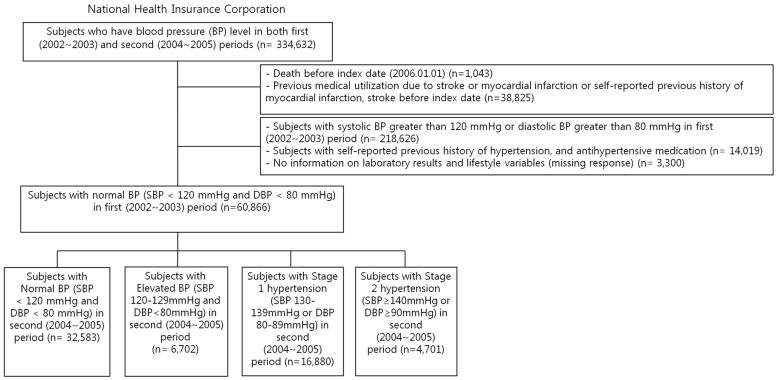
Flow chart of the study population.

**Figure 2 jcm-08-00820-f002:**
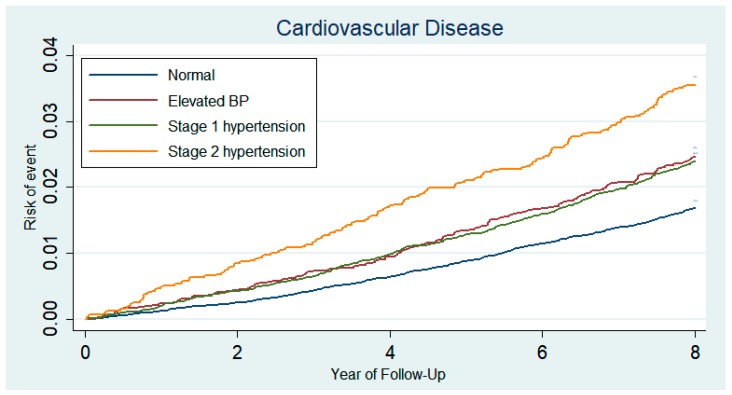
Kaplan–Meier curves demonstrating the incidence of cardiovascular disease (composite of nonfatal myocardial infarction, stroke, or cardiovascular death) according to the two-year blood pressure changes in people with previously normal blood pressure. Subjects had normal BP in first (2002 to 2003) periods and whose BP checked in second (2004–2005) periods was divided into four groups: normal (SBP < 120 mmHg and DBP < 80 mmHg), 2017 ACC/AHA defined Elevated BP (SBP 120–129 mmHg and DBP < 80 mmHg), Stage 1 hypertension (SBP 130–139 mmHg or DBP 80–89 mmHg) and Stage 2 hypertension (SBP ≥ 140 mmHg or DBP ≥ 90 mmHg).

**Table 1 jcm-08-00820-t001:** Descriptive characteristics of study participants on their second health examination.

Change in BP Group	Total (All)	Normal (SBP < 120 mmHg and DBP < 80 mmHg)	Elevated BP (SBP 120–129 mmHg and DBP < 80 mmHg)	Stage 1 Hypertension (SBP 130–139 mmHg or DBP 80–89 mmHg)				Stage 2 Hypertension (SBP ≥ 140 mmHg or DBP ≥ 90 mmHg)	*p* ^†^
				Total (Stage 1 Hypertension)	IDH (SBP < 130 mmHg and DBP 80–89 mmHg)	ISH (SBP 130–139 mmHg and DBP < 80 mmHg)	SDH (SBP 13–139 mmHg and DBP 80–89 mmHg)		
Number of people	60,866	32,583	6702	16,880	10,515	1825	4540	4701	
SBP, mmHg *	114.9 ± 12.8	106.0 ± 7.5	122.1 ± 2.8	123.2 ± 8.1	118.1 ± 5.6	132.1 ± 2.7	131.4 ± 2.6	137.1 ± 11.2	
	(108, 113, 120)	(100, 110, 110)	(120, 120, 124)	(120, 120, 130)	(117, 120, 120)	(130, 130, 134)	(130, 130, 132)	(130, 140, 141)	
DBP, mmHg *	72.4 ± 9.3	66.3 ± 6.0	71.1 ± 4.5	80.0 ± 3.7	80.7 ± 1.8	72.2 ± 4.4	81.5 ± 2.7	88.8 ± 6.8	
	(67, 70, 80)	(60, 69, 70)	(70, 70, 74)	(80, 80, 80)	(80, 80, 80)	(70, 71, 75)	(80, 80, 82)	(90, 90, 90)	
Age, Mean (SD) or %	51.5 (7.6)	50.9 (7.2)	52.0 (8.0)	51.9 (7.8)	51.0 (7.2)	54.4 (9.0)	52.7 (8.1)	53.5 (8.6)	<0.001
40–49	50.7	53.7	48.8	48.5	52.6	37.8	43.5	41.1	<0.001
50–59	33.9	33.3	33.1	34.7	34	33.8	36.7	36.1	
60–69	12.3	10.7	14.3	13.3	11	21.4	15.3	16.7	
70 or older	3.1	2.4	3.8	3.5	2.4	7.1	4.6	6.1	
Sex									<0.001
Female	52.5	59.3	51	43.7	42.8	51.2	42.7	38.4	
BMI, kg/m^2^, Mean (SD) or %	23.0 (2.6)	22.6 (2.5)	23.2 (2.6)	23.4 (2.6)	23.2 (2.6)	23.5 (2.6)	23.6 (2.7)	23.8 (2.9)	<0.001
<18.5	3.5	4.3	2.8	2.5	2.5	2.1	2.7	2.6	<0.001
18.5–22.9	48.7	53.8	46.2	43.1	45.2	40.1	39.4	37.3	
23.0–24.9	26.7	25.2	27.5	28.8	28.5	30.2	28.9	28.1	
≥25.0	21.1	16.7	23.5	25.6	23.8	27.6	29	32.1	
FSG, mg/dL, Mean (SD) or %	92.8 (22.8)	91.2 (20.0)	94.1 (25.6)	94.3 (25.2)	93.5 (25.3)	96.0 (27.2)	95.4 (24.1)	96.4 (25.9)	<0.001
<100.0	77.3	80.6	74.7	74.3	76.2	71.3	71.1	69.1	<0.001
100–125.9	19	16.6	21.1	21.1	19.8	22.6	23.3	24.5	
≥126.0	3.7	2.8	4.2	4.6	4	6.1	5.6	6.4	
Total cholesterol, mg/dL, Mean (SD) or %	193.6 (35.0)	191.5 (34.5)	195.1 (34.9)	195.7 (35.4)	194.9 (35.1)	196.9 (35.8)	197.0 (36.0)	198.6 (36.1)	<0.001
<200.0	59.7	62.1	58	57.3	58.3	54.4	56.2	54.3	<0.001
200.0–239.9	30.4	29.1	30.8	31.8	31.4	34.6	31.8	32.7	
≥240.0	9.9	8.7	11.2	10.9	10.4	11	12	13	
Physical activity, times per week, Mean (SD) or %	1.9 (1.1)	1.9 (1.1)	1.9 (1.2)	1.9 (1.1)	1.9 (1.1)	1.9 (1.2)	1.8 (1.1)	1.8 (1.1)	<0.001
None	50.6	50.7	49.1	50.3	50.1	51.1	50.4	53.5	<0.001
1–2	27.8	27.6	27.6	28.7	28.7	25.9	29.8	27	
3–4	12.3	12.5	12.7	12.2	12.8	12.7	10.7	10.9	
5–7	9.2	9.2	10.6	8.8	8.4	10.3	9.1	8.6	
Smoking status									<0.001
Never smoker	71.4	75.1	70.8	66.6	66	70.3	66.4	64.1	
Ex-smoker	8	6.7	9	9.4	9.3	9.6	9.6	10	
Current smoker	20.6	18.1	20.3	24	24.7	20.1	24	25.9	
Drinking habit, drinks per wk., Mean (SD) or %	1.7 (1.0)	1.6 (1.0)	1.7 (1.1)	1.8 (1.1)	1.8 (1.1)	1.7 (1.1)	1.8 (1.1)	1.9 (1.2)	<0.001
None	61.7	65.7	60.9	56.7	55.7	62.6	56.5	53.5	<0.001
1–2	31.1	29	31.4	34.2	35.3	29.8	33.4	34.5	
≥ 3	7.2	5.3	7.7	9.1	9	7.6	10	12	
SES, %									<0.001
Lower half	31.8	30.7	30.9	32.7	31	34.6	35.7	36.9	
Upper half	68	69.1	68.7	67.1	68.8	65.3	64	62.7	
Medical Aid	0.3	0.2	0.4	0.2	0.2	0.2	0.3	0.4	
CCI, Mean (SD) or %	1.1 (1.2)	1.1 (1.2)	1.1 (1.2)	1.1 (1.2)	1.1 (1.2)	1.1 (1.2)	1.1 (1.2)	1.1 (1.2)	0.034
0	37.3	36.5	37.1	38.3	39.2	35.3	37.5	38.7	<0.001
1–2	51.6	52.2	51.8	50.8	50.2	53.5	50.8	50.3	
≥ 3	11.1	11.2	11.1	10.9	10.6	11.2	11.6	11	
Aspirin, %									0.001
Yes	1.5	1.4	1.8	1.5	1.3	1.6	1.9	2	
Statin, %									0.003
Yes	2.3	2.1	2.7	2.4	2.4	3	2.3	2	

* Values are mean ± standard deviation and (25th percentile, 50th percentile, 75th percentile) ^†^
*p* Values are those of analysis of variance (ANOVA) test for continuous variables or chi-square test for categorical variables. Acronyms: SBP, Systolic Blood Pressure; DBP, Diastolic Blood Pressure, IDH, isolated diastolic hypertension; ISH, isolated systolic hypertension; SDH, combined systolic/diastolic hypertension; SD, Standard Deviation; BMI, body mass index; FSG, fasting serum glucose; SES, socioeconomic status; CCI, Charlson comorbidity index.

**Table 2 jcm-08-00820-t002:** Associations between change in blood pressure according to the 2017 ACC/AHA high blood pressure guideline and risk of cardiovascular disease, myocardial infarction, stroke, and mortality.

Change in BP Group	Normal (SBP < 120 mmHg and DBP < 80 mmHg)	Elevated BP (SBP 120–129 mmHg and DBP < 80 mmHg)	Stage 1 Hypertension (SBP 130–139 mmHg or DBP 80–89 mmHg)	Stage 2 Hypertension (SBP ≥ 140 mmHg or DBP ≥ 90 mmHg)	*p* for Trend
Cardiovascular disease, *N* (%)	548 (1.68)	166 (2.48)	404 (2.39)	167 (3.55)	
Model 1	Reference	1.28	1.25	1.58	<0.001
95% CI		1.07–1.52	1.10–1.42	1.33–1.88	
Model 2	Reference	1.26	1.23	1.52	<0.001
95% CI		1.06–1.50	1.08–1.40	1.27–1.82	
Myocardial Infarction, *N* (%)	54 (0.17)	26 (0.39)	58 (0.34)	18 (0.38)	
Model 1	Reference	1.95	1.6	1.56	0.017
95% CI		1.22–3.12	1.10–2.32	0.91–2.67	
Model 2	Reference	1.91	1.54	1.5	0.033
95% CI		1.19–3.06	1.06–2.25	0.87–2.58	
Ischemic stroke, *N* (%)	181 (0.56)	64 (0.95)	154 (0.91)	71 (1.51)	
Model 1	Reference	1.41	1.37	1.86	<0.001
95% CI		1.06–1.88	1.10–1.70	1.41–2.45	
Model 2	Reference	1.38	1.32	1.73	<0.001
95% CI		1.04–1.84	1.06–1.64	1.30–2.29	
Hemorrhagic stroke, *N* (%)	82 (0.25)	25 (0.37)	57 (0.34)	34 (0.72)	
Model 1	Reference	1.31	1.21	2.23	0.002
95% CI		0.84–2.06	0.86–1.69	1.49–3.35	
Model 2	Reference	1.34	1.23	2.26	0.002
95% CI		0.85–2.10	0.87–1.73	1.49–3.41	
All–cause mortality, *N* (%)	683 (2.10)	176 (2.63)	436 (2.58)	222 (4.72)	
Model 1	Reference	0.99	0.97	1.43	0.006
95% CI		0.84–1.17	0.86–1.10	1.23–1.66	
Model 2	Reference	1.02	1	1.46	0.002
95% CI		0.86–1.21	0.89–1.13	1.25–1.70	
CVD mortality, *N* (%)	66 (0.20)	24 (0.36)	47 (0.28)	28 (0.60)	
Model 1	Reference	1.31	1.05	1.68	0.104
95% CI		0.82–2.09	0.72–1.54	1.07–2.62	
Model 2	Reference	1.37	1.1	1.74	0.072
95% CI		0.86–2.20	0.75–1.60	1.11–2.74	

Model 1: hazard ratio adjusted for age, and sex; Model 2: additionally adjusted for BMI, fasting serum glucose, total cholesterol, physical activity, smoking status, drinking habit, socioeconomic status, Charlson comorbidity index, Aspirin, and Statin medication. Acronyms: SBP, Systolic Blood Pressure; DBP, Diastolic Blood Pressure, CVD, Cardiovascular disease; 2017 ACC/AHA high blood pressure guideline, 2017 ACC/AHA/AAPA/ABC/ACPM/AGS/APhA/ASH/ASPC/NMA/PCNA Guideline for the Prevention, Detection, Evaluation, and Management of High Blood Pressure in Adults.

**Table 3 jcm-08-00820-t003:** Additional analysis of associations between change in blood pressure according to the 2017 ACC/AHA high blood pressure guideline and risk of cardiovascular disease, myocardial infarction, stroke, and mortality.

Change in BP Group	Normal (SBP < 120 mmHg and DBP < 80 mmHg)	Elevated BP (SBP 120–129 mmHg and DBP < 80 mmHg)	Stage 1 Hypertension(SBP 130–139 mmHg or DBP 80–89 mmHg)			Stage 2 Hypertension (SBP ≥ 140 mmHg or DBP ≥ 90 mmHg)	*p* for Trend
			IDH (SBP < 130 mmHg and DBP 80–89 mmHg)	ISH (SBP 130–139 mmHg and DBP < 80 mmHg)	SDH (SBP 130–139 mmHg and DBP 80–89 mmHg)		
Cardiovascular disease, *N* (%)	548 (1.68)	166 (2.48)	208 (1.98)	59 (3.23)	137 (3.02)	167 (3.55)	
Model 1	Reference	1.28	1.12	1.34	1.45	1.58	<0.001
95% CI		1.08–1.52	0.96–1.32	1.03–1.76	1.2–1.75	1.33–1.88	
Model 2	Reference	1.26	1.12	1.31	1.42	1.52	<0.001
95% CI		1.06–1.50	0.95–1.31	1.00–1.72	1.17–1.71	1.28–1.82	
Myocardial Infarction, *N* (%)	54 (0.17)	26 (0.39)	34 (0.32)	4 (0.22)	20 (0.44)	18 (0.38)	
Model 1	Reference	1.95	1.58	0.95	1.91	1.56	0.03
95% CI		1.22–3.12	1.02–2.42	0.34–2.62	1.14–3.2	0.91–2.67	
Model 2	Reference	1.91	1.52	0.91	1.87	1.5	0.054
95% CI		1.19–3.06	0.98–2.34	0.33–2.52	1.11–3.15	0.87–2.58	
Ischemic stroke, *N* (%)	181 (0.56)	64 (0.95)	73 (0.69)	28 (1.53)	53 (1.17)	71 (1.51)	
Model 1	Reference	1.42	1.16	1.72	1.57	1.86	<0.001
95% CI		1.06–1.88	0.89–1.53	1.15–2.57	1.16–2.14	1.41–2.46	
Model 2	Reference	1.38	1.14	1.63	1.51	1.73	<0.001
95% CI		1.04–1.84	0.87–1.5	1.09–2.44	1.10–2.06	1.31–2.29	
Hemorrhagic stroke, *N* (%)	82 (0.25)	25 (0.37)	31 (0.29)	11 (0.60)	15 (0.33)	34 (0.72)	
Model 1	Reference	1.31	1.14	1.73	1.09	2.24	0.001
95% CI		0.84–2.06	0.76–1.73	0.92–3.25	0.63–1.89	1.49–3.36	
Model 2	Reference	1.34	1.17	1.76	1.1	2.26	0.001
95% CI		0.86–2.11	0.77–1.77	0.93–3.33	0.63–1.92	1.49–3.41	
All-cause mortality, *N* (%)	683 (2.10)	176 (2.63)	227 (2.16)	75 (4.11)	134 (2.95)	222 (4.72)	
Model 1	Reference	0.99	0.92	1.15	0.99	1.43	0.001
95% CI		0.84–1.17	0.79–1.06	0.9–1.47	0.82–1.2	1.23–1.67	
Model 2	Reference	1.02	0.94	1.26	1.01	1.46	<0.001
95% CI		0.86–1.21	0.81–1.09	0.99–1.60	0.84–1.22	1.25–1.71	
CVD mortality, *N* (%)	66 (0.20)	24 (0.36)	14 (0.13)	12 (0.66)	21 (0.46)	28 (0.60)	
Model 1	Reference	1.32	0.6	1.65	1.51	1.68	0.017
95% CI		0.82–2.10	0.34–1.07	0.89–3.07	0.92–2.47	1.08–2.63	
Model 2	Reference	1.38	0.62	1.79	1.57	1.75	0.011
95% CI		0.86–2.21	0.35–1.11	0.96–3.34	0.95–2.59	1.11–2.75	

Model 1: hazard ratio adjusted for age, and sex; Model 2: additionally adjusted for BMI, fasting serum glucose, total cholesterol, physical activity, smoking status, drinking habit, socioeconomic status, Charlson comorbidity index, Aspirin, and Statin medication. Acronyms: SBP, Systolic Blood Pressure; DBP, Diastolic Blood Pressure, CVD, Cardiovascular disease; IDH, isolated diastolic hypertension; ISH, isolated systolic hypertension; SDH, combined systolic/diastolic hypertension; 2017 ACC/AHA high blood pressure guideline, 2017 ACC/AHA/AAPA/ABC/ACPM/AGS/APhA/ASH/ASPC/NMA/PCNA Guideline for the Prevention, Detection, Evaluation, and Management of High Blood Pressure in Adults.

**Table 4 jcm-08-00820-t004:** Additional analysis of associations between three categorical changes in blood pressure and risk of cardiovascular disease in 2017 ACC/AHA defined Stage 1 hypertension.

Change in BP Group	IDH (SBP < 130 mmHg and DBP 80–89 mmHg)	ISH (SBP 130–139 mmHg and DBP < 80 mmHg)	SDH (SBP 130–139 mmHg and DBP 80–89 mmHg)
Number of people	10,515	1825	4540
SBP, mmHg *	118.1 ± 5.6 (117, 120, 120)	132.1 ± 2.7 (130, 130, 134)	131.4 ± 2.6 (130, 130, 132)
DBP, mmHg *	80.7 ± 1.8 (80, 80, 80)	72.2 ± 4.4 (70, 71, 75)	81.5 ± 2.7 (80, 80, 82)
Cardiovascular disease, *N* (%)	208 (1.98)	59 (3.23)	137 (3.02)
aHR (95% CI) †	Reference	1.23 (0.92–1.66)	1.29 (1.04–1.61)

* Values are mean ± standard deviation and (25th percentile, 50th percentile, 75th percentile); ^†^ adjusted hazard ratio (aHR) and 95% confidence interval (CI): adjustment for age, BMI, fasting serum glucose, total cholesterol, physical activity, smoking status, drinking habit, socioeconomic status, Charlson comorbidity index, Aspirin, and Statin medication. Acronyms: SBP, Systolic Blood Pressure; DBP, Diastolic Blood Pressure, CVD, Cardiovascular disease; IDH, isolated diastolic hypertension; ISH, isolated systolic hypertension; SDH, combined systolic/diastolic hypertension; 2017 ACC/AHA, 2017 ACC/AHA/AAPA/ABC/ACPM/AGS/APhA/ASH/ASPC/NMA/PCNA Guideline for the Prevention, Detection, Evaluation, and Management of High Blood Pressure in Adults.

**Table 5 jcm-08-00820-t005:** Subgroup analysis of associations between change in blood pressure and risk of cardiovascular disease *.

Change in BP Group	Normal (SBP < 120 mmHg and DBP < 80 mmHg)	Elevated BP (SBP 120–129 mmHg and DBP < 80 mmHg)	Stage 1 Hypertension (SBP 130–139 mmHg or DBP 80–89 mmHg)			Stage 2 Hypertension (SBP ≥ 140 mmHg or DBP ≥ 90 mmHg)	*p* for Trend	*p* for Interaction
			IDH (SBP < 130 mmHg and DBP 80–89 mmHg)	ISH (SBP 130–139 mmHg and DBP < 80 mmHg)	SDH (SBP 130–139 mmHg and DBP 80–89 mmHg)			
Age								0.339
40–64	Reference	1.37 (1.11–1.69)	1.17 (0.97–1.41)	1.22 (0.82–1.81)	1.62 (1.29–2.03)	1.66 (1.33–2.08)	<0.001	
65 or older	Reference	1.13 (0.82–1.54)	0.96 (0.70–1.33)	1.52 (1.04–2.23)	1.17 (0.83–1.64)	1.44 (1.08–1.93)	0.012	
Sex								0.98
Male	Reference	1.33 (1.05–1.68)	1.14 (0.93–1.41)	1.08 (0.72–1.62)	1.53 (1.21–1.96)	1.48 (1.17–1.86)	<0.001	
Female	Reference	1.20 (0.92–1.56)	1.07 (0.83–1.38)	1.60 (1.11–2.31)	1.25 (0.91–1.71)	1.60 (1.21–2.11)	0.001	
FSG, mg/dL, %								0.425
<126.0	Reference	1.24 (1.04–1.50)	1.15 (0.98–1.36)	1.41 (1.06–1.87)	1.39 (1.14–1.70)	1.56 (1.30–1.88)	<0.001	
≥126.0	Reference	1.45 (0.83–2.55)	0.68 (0.34–1.34)	0.61 (0.22–1.75)	1.74 (0.98–3.10)	1.32 (0.72–2.42)	0.282	
Smoking status								0.683
Never smoker	Reference	1.27 (1.02–1.59)	1.13 (0.92–1.38)	1.47 (1.06–2.04)	1.20 (0.92–1.55)	1.63 (1.31–2.04)	<0.001	
Ex- or Current smoker	Reference	1.23 (0.92–1.64)	1.08 (0.83–1.39)	1.04 (0.64–1.70)	1.75 (1.32–2.32)	1.32 (0.98–1.78)	0.003	
CCI								0.23
0–1	Reference	1.32 (1.05–1.66)	1.15 (0.93–1.42)	1.53 (1.09–2.15)	1.45 (1.13–1.86)	1.62 (1.30–2.04)	<0.001	
≥2	Reference	1.19 (0.90–1.56)	1.07 (0.83–1.38)	0.99 (0.63–1.56)	1.36 (1.01–1.83)	1.37 (1.02–1.83)	0.017	

* Values are adjusted hazard ratio and 95% confidence interval, adjustment for age, sex, BMI, fasting serum glucose, total cholesterol, physical activity, smoking status, drinking habit, socioeconomic status, Charlson comorbidity index, Aspirin, and Statin medication. Acronyms: SBP, Systolic Blood Pressure; DBP, Diastolic Blood Pressure, CVD, Cardiovascular disease; IDH, isolated diastolic hypertension; ISH, isolated systolic hypertension; SDH, combined systolic/diastolic hypertension.

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
