# Peer review of "Blood Pressure Change from Normal to 2017 ACC/AHA Defined Stage 1 Hypertension and Cardiovascular Risk"

_jcm, 2019, doi:10.3390/jcm8060820_

Reviewer 1 Report

The new ACC / AHA blood pressure guidelines are strongly discussed in the field. This manuscript by Son et al could bring additional insights for the risk of subjects, which have been diagnosed with normal blood pressure up to now.

While the presentation of the study and the results is overall good, some questions remain:

Why were age and BMI divided in categories and not used as continuous variables in the statistical models?

Please refer to Table 4 in the results section, not only in the discussion.

Numbers are given as mean and standard deviation. However, this might not be appropriate in all cases due to non-normality of the underlying distribution. For example, DBP is limited to 80-89 in IDH patients, mean value is 80.7 and SD is 1.8, thus 1 SD is already outside of the possible range of values. Please check whether an alternative representation (e.g. median and quartiles) is more appropriate.

Table 2 and Table 3 seem to contain the same numbers for Elevated BP and Stage 2 Hypertension. However, some numbers disagree slightly, e.g. hazard ratio for elevated BP, model 1, CVD Mortality is 1.31 in Table 2 and 1.32 in Table 3. Can you explain?

The legend of figure 2 could be shortened.

Author Response

We would like to thank the reviewer for this excellent comment.

The author's reply to the review report is in the attached file.

Response to Reviewer 1 Comments

Point 1: Why were age and BMI divided in categories and not used as continuous variables in the statistical models? 

Response 1:

We would like to thank the reviewer for this excellent comment.

As you know, it is necessary to assume that the continuous variable affects the outcome variable constantly in order to adjust the continuous variable in the multivariate analysis.

However, in the preliminary analysis, the effect of age and BMI on outcome variables varied with ranges.[For example; Age 40-49 reference, Age 50-59 (aHR 1.60; 95% CI, 1.38-1.86), Age 60-69 (aHR 3.61; 95% CI, 3.08-4.23), Age 70 or older (aHR 8.31; 95% CI, 6.91-10.00)]

So age and BMI were used as categories in the statistical model.

Point 2: Please refer to Table 4 in the results section, not only in the discussion.

Response 2:

I added it to the result section as follows.

Page 5, lines 151-153:

'Compared to those with stage 1 IDH, people with stage 1 SDH had higher CVD risk (aHR 1.29, 95% CI, 1.04-1.61) (Table 4).'

Point 3: Numbers are given as mean and standard deviation. However, this might not be appropriate in all cases due to non-normality of the underlying distribution. For example, DBP is limited to 80-89 in IDH patients, mean value is 80.7 and SD is 1.8, thus 1 SD is already outside of the possible range of values. Please check whether an alternative representation (e.g. median and quartiles) is more appropriate.

Response 3:

I would like to thank the reviewer for this excellent comment.

I added 25th percentile, median, and 75th percentile of blood pressure to table 1 and table 4 to help readers understand it properly.

Point 4: Table 2 and Table 3 seem to contain the same numbers for Elevated BP and Stage 2 Hypertension. However, some numbers disagree slightly, e.g. hazard ratio for elevated BP, model 1, CVD Mortality is 1.31 in Table 2 and 1.32 in Table 3. Can you explain?

Response 4:

As you know, in a regression analysis, the coefficient values of the adjusting variables can be changed in the process of increasing the fitness of the model according to the number of groups analyzed.

I did a comparison between the 4 groups in Table 2, and a comparison between the 6 groups in Table 3, so there might be slight differences in some numbers.

Point 5: The legend of figure 2 could be shortened.

Response 5:

I have reduced the legend of figure 2 according to your recommendation.

Page 12, lines 188-191:

Figure 2. Kaplan-Meier curves demonstrating the incidence of cardiovascular disease (composite of nonfatal myocardial infarction, stroke, or cardiovascular death) according to the two-year blood pressure changes in people with previously normal blood pressure. Subjects had normal BP in first (2002 to 2003) periods and whose BP checked in second (2004-2005) periods was divided into four groups: normal (SBP < 120 mmHg and DBP < 80 mmHg), 2017 ACC/AHA defined elevated BP (SBP 120-129 mmHg and DBP < 80 mmHg), stage 1 hypertension (SBP 130-139 mmHg or DBP 80-89 mmHg) and Stage 2 hypertension (SBP ≥ 140 mmHg or DBP ≥ 90 mmHg).

Reviewer 2 Report

Joung Sik Son et al. the cv risk of patients progressing from normal BP to stage 1 hypertension over a time period of 2 years. The patients were the followed for almost 8 years.

The study question is a very interesting one: does the progression from one BP category to another itself increase the risk of CVD. However, unfortunately, this question cannot be answered with this manuscript. Considering that cv risk associated with BP follows a J-curve, the higher BP during the second period should increase the cv risk, otherwise, the ACC/AHA guidelines would be completely wrong in their assessment that stage 1 hypertension is a thing and needs treatment. The better way to examine this would have been to take all patients with a stage 1 hypertension during the second period and compare those with normal vs those with stage 1 hypertension during the first period. However we can see a risk change in those patients with stage 1 hypertension -whether that is from the stage of hypertension or the change over time cannot be answered in my eyes.

If you want to stick to the data you have you need to rephrase your interpretation of the results (otherwise, if available, support your interpretation with new data as discussed above).

Minor comments:

- Abstract, line 17-18: Sentence is difficult to understand

- Materials and methods, line 94-97: Sentence is difficult to understand

- Materials and methods, line 108-110: A reference needs to be added

- please do not repeat newly defined each time you state ACC/AHA guidelines. Consider using a clear abbreviation

- Tables: I am not a statistician, so please check with a statistician:

     - for the p-value calculated with chi2: which groups did you use? Please mark clearly.

     - Table 1: sex: male/female are mutually exclusive (at least do your numbers not suggest that there were any non-binary genders involved) - please state only one.

     - Table 2: Model 2 seems to involve a lot of parameters used in a multivariate calculation considering the relatively small number of outcomes: please check with a statistician if appropriate. The value is rather called Reference than Referent

- Discussion, line 236-239: There is data available that even in very normal blood pressure value a short-term masked hypertension could be hidden (Burkard T et al. Heart 2018;104:1173-1179). Please add to your discussion of limitations.

- Discussion, line 241-243: I cannot understand this sentence.

- Reference 5: The author should not be Collaboration, P.S. but Lewington S et al.

Author Response

We would like to thank the reviewer for this excellent comment.

The author's reply to the review report is in the attached file.

Response to Reviewer 2 Comments

Point 1: Joung Sik Son et al. the cv risk of patients progressing from normal BP to stage 1 hypertension over a time period of 2 years. The patients were the followed for almost 8 years.

The study question is a very interesting one: does the progression from one BP category to another itself increase the risk of CVD. However, unfortunately, this question cannot be answered with this manuscript. Considering that cv risk associated with BP follows a J-curve, the higher BP during the second period should increase the cv risk, otherwise, the ACC/AHA guidelines would be completely wrong in their assessment that stage 1 hypertension is a thing and needs treatment. The better way to examine this would have been to take all patients with a stage 1 hypertension during the second period and compare those with normal vs those with stage 1 hypertension during the first period. However we can see a risk change in those patients with stage 1 hypertension -whether that is from the stage of hypertension or the change over time cannot be answered in my eyes.

If you want to stick to the data you have you need to rephrase your interpretation of the results (otherwise, if available, support your interpretation with new data as discussed above).

Response 1:

We would like to thank the reviewer for this excellent comment, which has helped us clarify an important aspect of our study design that was previously unclear.

As you said, the linear correlation between blood pressure and the risk of developing CVD is well known.

However, we did this study with an interest in the following two points.

First, we wanted to identify the relationship between changes in BP for about two years and risk of CVD in Korean adults.

Since many Koreans receive a biennial national health screening, it was necessary to study the clinical significance of BP changes for two years.

Second, we hoped to analyze the clinical implications of BP changes according to the 2017 ACC/AHA hypertension guideline BP categories.

2017 ACC/AHA Stage 1 hypertension was defined as SBP 130-139mmHg or DBP 80-89mmHg. Therefore, the participants was diagnosed as hypertension if DBP was 80 mmHg in a national health screening. This caused many people with normal BP in a previous health screening to be diagnosed with hypertension in the next health checkup. It was therefore necessary to assess the risk of CVD among these people.

In the preliminary analysis, we also found that there was a heterogeneity in the risk of CVD in Stage 1 hypertension. Therefore, we analyzed the Stage 1 hypertension by dividing into the following three groups; Stage 1 isolated diastolic hypertension (IDH) (SBP<130 DBP 80-89 mmHg), Stage 1 isolated systolic hypertension (ISH) (SBP 130-139 DBP<80 mmHg), and Stage 1 combined systolic/diastolic hypertension (SDH) (SBP 130-139 DBP 80-89 mmHg) groups.

We found that participants with BP elevation from normal BP to Stage 1 hypertension had higher risk for CVD (aHR 1.23; 95% CI, 1.08-1.40) but BP elevation to Stage 1 IDH (SBP<130 and DBP 80-89 mmHg) was not significantly related with CVD risk (aHR 1.12; 95%CI, 0.95-1.31).

We believed that these findings are important for public health and for many people who have been diagnosed with Stage 1 hypertension, even though they were normal BP in the previous health checkup.

Because we conducted the research with these two purposes, we used the study subjects and methods used in this study.

Point 2: Abstract, line 17-18: Sentence is difficult to understand.

Response 2:

For clarity, the sentences have been modified as follows.

Page 1, lines 17-18:

Study subjects had normal BP (SBP<120mmHg and DBP<80mmHg), no history of anti-hypertensive medication, and cardiovascular disease (CVD) in first period (2002-2003). The BP change was defined according to the BP difference between the first and second period (2004-2005).

Point 3: Materials and methods, line 94-97: Sentence is difficult to understand.

Response 3:

For clarity, the sentences have been modified as follows.

Page 4, lines 94-99:

The primary outcome was CVD. A CVD event was defined as a composite end point of 2 or more days of hospitalization due to acute myocardial infarction (AMI) (I21), stroke (I60-I69) or CVD mortality (I00-I99) according to the International Classification of Diseases, Tenth Revision (ICD-10) codes. The secondary outcomes were AMI (I21), ischemic stroke (I63), hemorrhagic stroke (I60-I62), all-cause mortality and CVD mortality.

Point 4: Materials and methods, line 108-110: A reference needs to be added.

Response 4:

DDD is defined by the WHO as the drug units representing dosages with approximately similar efficacy.

As you suggested, I added references.

Point 5: please do not repeat newly defined each time you state ACC/AHA guidelines. Consider using a clear abbreviation.

Response 5:

'2017 ACC/AHA defined Stage 1 hypertension' is denoted as 'Stage 1 hypertension' and '2017 ACC/AHA defined elevated BP' is denoted as 'Elevated BP'.

Point 6: Tables: for the p-value calculated with chi2: which groups did you use? Please mark clearly.

Response 6:

P values are those of analysis of variance (ANOVA) test for continuous variables or chi-square test for categorical variables.

I have further clarified the P values and marked them in Table 1.

Point 7: Table 1: sex: male/female are mutually exclusive (at least do your numbers not suggest that there were any non-binary genders involved) - please state only one.

Response 7:

I revised to show only female in Table 1.

Point 8: Table 2: Model 2 seems to involve a lot of parameters used in a multivariate calculation considering the relatively small number of outcomes: please check with a statistician if appropriate. The value is rather called Reference than Referent.

Response 8:

The smallest outcome was 18 myocardial infarction cases in the Stage 2 Hypertension group, and the other outcomes were more frequent.

I contacted a statistics specialist and he responded that there was no statistically significant problem.

Referent has been replaced by Reference.

Point 9: Discussion, line 236-239: There is data available that even in very normal blood pressure value a short-term masked hypertension could be hidden (Burkard T et al. Heart 2018;104:1173-1179). Please add to your discussion of limitations.

Response 9:

I would like to thank the reviewer for this excellent comment.

I added the research results that you recommended to the discussion section.

Page 13, lines 240-242:

A cross-sectional study showed that there were significant differences between a single office BP measurement and the mean of consecutive BP measurements.

Therefore, a short-term masked hypertension might not be found using only one BP measurement value.

Point 10: Discussion, line 241-243: I cannot understand this sentence.

Response 10:

I removed the unclear sentences.

Because the incidence of acute myocardial infarction in men and women was not separately analyzed, the sentence was deleted.

Page 13-14, lines 244-247:

Further studies are needed to analyze the relationship between drug use during the follow-up period and changes in BP and the incidence of CVD.

Third, the study subjects were Koreans. Therefore, there is a limit to generalize research results to other countries or races.

Point 11: Reference 5: The author should not be Collaboration, P.S. but Lewington S et al.

Response 11:

I corrected it as follows.

Lewington S, Clarke R, Qizilbash N, Peto R, Collins R. Age-specific relevance of usual blood pressure to vascular mortality: a meta-analysis of individual data for one million adults in 61 prospective studies. The Lancet 2002, 360, 1903-1913.

Round  2

Reviewer 2 Report

All comments have been adressed, thank you.